# SARS-CoV-2 Invasion and Pathological Links to Prion Disease

**DOI:** 10.3390/biom12091253

**Published:** 2022-09-07

**Authors:** Walter J. Lukiw, Vivian R. Jaber, Aileen I. Pogue, Yuhai Zhao

**Affiliations:** 1LSU Neuroscience Center, Louisiana State University Health Science Center, New Orleans, LA 70112, USA; 2Alchem Biotek Research, Toronto, ON M5S 1A8, Canada; 3Department of Ophthalmology, LSU Health Science Center, New Orleans, LA 70112, USA; 4Department Neurology, LSU Health Science Center, New Orleans, LA 70112, USA; 5Department of Cell Biology & Anatomy, LSU Health Science Center, New Orleans, LA 70112, USA

**Keywords:** angiotensin-converting enzyme 2 receptor (ACE2R), Creutzfeldt–Jakob disease (CJD), cytokine storm, microRNA (miRNA), miRNA-146a, miRNA-155, prion disease (PrD), SARS-CoV-2

## Abstract

Severe acute respiratory syndrome coronavirus 2 (SARS-CoV-2), the causative agent of the COVID-19 disease, is a highly infectious and transmissible viral pathogen that continues to impact human health globally. Nearly ~600 million people have been infected with SARS-CoV-2, and about half exhibit some degree of continuing health complication, generically referred to as ***long COVID***. Lingering and often serious neurological problems for patients in the post-COVID-19 recovery period include ***brain fog***, behavioral changes, confusion, delirium, deficits in intellect, cognition and memory issues, loss of balance and coordination, problems with vision, visual processing and hallucinations, encephalopathy, encephalitis, neurovascular or cerebrovascular insufficiency, and/or impaired consciousness. Depending upon the patient’s age at the onset of COVID-19 and other factors, up to ~35% of all elderly COVID-19 patients develop a mild-to-severe encephalopathy due to complications arising from a SARS-CoV-2-induced ***cytokine storm*** and a surge in cytokine-mediated pro-inflammatory and immune signaling. In fact, this ***cytokine storm syndrome:*** **(i)** appears to predispose aged COVID-19 patients to the development of other neurological complications, especially those who have experienced a more serious grade of COVID-19 infection; **(ii)** lies along highly interactive and pathological pathways involving SARS-CoV-2 infection that promotes the parallel development and/or intensification of progressive and often lethal neurological conditions, and **(iii)** is strongly associated with the symptomology, onset, and development of human prion disease (PrD) and other insidious and incurable neurological syndromes. This commentary paper will evaluate some recent peer-reviewed studies in this intriguing area of human SARS-CoV-2-associated neuropathology and will assess how chronic, viral-mediated changes to the brain and CNS contribute to cognitive decline in PrD and other progressive, age-related neurodegenerative disorders.

## 1. Introduction

Viral and other microbial infections of the human nervous tissues have long been recognized for their ability to initiate, propagate, or intensify the same neuropathological, inflammatory, and/or degenerative changes that are observed over the entire continuum of progressive, neurodegenerative brain diseases, which include Creutzfeldt–Jakob disease (CJD) and other PrDs [1,2,3,4,5,6,7,8,9]. Multiple independent and peer-reviewed published reports indicate that both DNA and RNA viruses, such as the human double-stranded DNA (dsDNA) *Herpes simplex* type 1 and 2 (HSV-1, HSV-2), the human cytomegalovirus (HMCV), the Epstein–Barr virus (EBV), ssRNA viruses such as the hepatitis C virus (HCV; *Herpesviridae*), human influenza A viruses (H1N1/H3N2; *Orthomyxoviridae*), Zika virus (ZIKVs; *Flaviviridae*), MERS-CoV (*Coronaviridae*), severe acute respiratory syndrome coronavirus 2 (SARS-CoV-2; *Coronaviridae*), and a remarkably large number of bacteria from the genera *Aggregatibacter*, *Bacteroides*, *Borrelia*, *Chlamydia*, *Eikenella*, *Fusobacterium*, *Porphyromonas*, *Prevo-tella*, *Tannerella*, *and Treponema*, as well as several other fungi (*Aspergillus*; *Candida*) or eukaryotic parasites (e.g., *Toxicara*; *Toxoplasma*) have been associated with an acceleration of age-dependent inflammatory neurodegeneration [2,3,4,5,6,7,8,9,10,11,12,13,14]. More importantly, all microbial infections of the brain and CNS contribute to the development of a microbial- and/or viral-induced ***cytokine storm***, which is associated with a progressive inflammatory degeneration of brain cells and tissues. The continuing progression of this ***cytokine storm*** is often accompanied by the appearance of amyloid-beta (Aβ) peptides, Aβ amyloid fibers, prion amyloids, related amyloidogenic, lipoprotein or Aβ peptide aggregation processes, the appearance of twisted neurofilamentous structures, including neurofibrillary tangles, pro-inflammatory biomarkers, gliosis of microglial cells, the formation of vacuoles and spongiform change within brain cells, neuronal cell loss, or any combination of these pathological biomarkers associated with a diverse range of human neurological disorders and phenotypic states [4,7,8,10,11,12,13,14,15,16,17,18,19,20].

The viral infection of human host cells often involves the direct recognition, binding, and interaction of the virus particle with a naturally occurring host cell surface receptor that protrudes from cholesterol-enriched lipid raft domains in the viral lipoprotein envelope (see Section 2. **SARS-CoV-2—Structure, Function and Neuroinvasion**) [21,22,23,24,25,26,27,28]. In the case of COVID-19, the SARS-CoV-2 ‘S1’ spike protein interacts specifically with the angiotensin-converting enzyme 2 receptor (ACE2R; EC 3.4.17.23; https://www.genecards.org/cgi-bin/carddisp.pl?gene=ACE2; last accessed on 30 August 2022), which is located on multiple host cell surfaces. The ACE2R: **(i)** is a ubiquitously expressed zinc-containing metallo-carboxypeptidase surface receptor glycoprotein of the human renin-angiotensin system (RAS) that has a natural homeostatic role in the regulation of blood pressure; **(ii)** is abundantly expressed in multiple endothelial and epithelial cells of the human respiratory tract; and **(iii)** has been identified on the outer surface of every human host cell type so far analyzed, except for erythrocytes [6,25,29,30,31]. With regard to neuroinvasion and neurotropism, the ACE2R has been abundantly detected in every cell type of the brain and central nervous system (CNS) that has been analyzed, in the neurovasculature, within the choroid plexus as well as along multiple visual processing and neuro-ophthalmic signaling tracts extending from the human retina into the primary visual areas of the occipital lobe [6,26,29,30,31,32]. The highest ACE2R densities in the human CNS have been localized to the brain’s medullary centers that include respiratory neurons of the medulla oblongata and pons in the brainstem, and this may partly explain the vulnerability of many SARS-CoV-2-infected patients to severe respiratory disturbances [6,30,33]. The remarkable ubiquity of the ACE2R indicates that the SARS-CoV-2 virus has an enormous potential to infect, damage, and/or destroy virtually every cell, tissue type, and organ system within the human host and to induce a serious and highly interactive multi-organ system failure, especially over the long term [34,35,36,37]. 

## 2. SARS-CoV-2: Structure, Function, and Neuroinvasion

The SARS-CoV-2 virus consists of a ~100 nm diameter virion particle containing an unusually large, positive-sense single-stranded RNA (ssRNA) genome of about 29,903 nucleotides packaged into a nucleocapsid core within a compact spherical lipoprotein envelope [27,38,39,40,41,42] (National Center for Biological Information (NCBI) GenBank Accession No. NC_045512.2; last accessed on 30 August 2022). Decorating the surface of this SARS-CoV-2 lipoprotein envelope are hundreds of homotrimeric, ~681-amino-acid, ~78.3 kDa ‘S1’ spike glycoproteins that play essential roles in the molecular mechanism of a successful SARS-CoV-2 invasion [38,42,43]. These include the initial recognition of the ACE2R receptor on susceptible host cells, SARS-CoV-2 attachment, as well as fusion with and entry into host cells to initiate SARS-CoV-2 infection [24,38,43]. The ‘S1’ spike protein of SARS-CoV-2 is an absolute requirement for ACE2R recognition and viral entry, and it is the main antigen used as a target for COVID-19 vaccines [25,29,44]. Vaccine-mediated blocking of ‘S1’ spike protein-ACE2R recognition in host cells fully arrests the initiation of SARS-CoV-2 infection [45,46]. Interestingly, the amino acid sequence of the SARS-CoV-2 viral ‘S1’ spike protein shares a number of ‘prion-like’ attributes and properties that appear to contribute to various aspects of the PrD neuropathology, neurophysiology, and the pathomechanism of prion-type infection and neurodegeneration. These attributes include the ability to bind various natural glycosoaminoglycans such as heparin, heparin binding proteins (HBP), and disease-associated molecules such as amyloid-beta (Aβ) peptides and prion proteins; the ‘S1’ spike protein therefore acts as ‘seeding centers’ for the formation of disease-characteristic intracellular inclusions in the brain and CNS. These pathological lesions support pro-inflammatory neurodegeneration, neuronal cell atrophy, death, and/or PrD-type change (see Section 4. **The SARS-CoV-2 and PrD—Overlapping Pathological Neurobiology**) [3,43,44,47,48,49]. 

## 3. Prion Disease (PrD) and Prion Neurobiology

PrDs of humans, also known as transmissible spongiform encephalopathies or TSEs, represent an expanding spectrum of progressive and ultimately lethal neurodegenerative disorders that globally affect about one person out of every one million per year [17,20,49]. About 85–90% of all PrDs manifest as the Creutzfeldt–Jakob disease (CJD) or a variant of the CJD (vCJD), with the remainder consisting mainly of the Gerstmann–Straussler–Scheinker (GSS) syndrome, fatal familial insomnia (FFI), variably protease-sensitive prionopathy (vPSPr), and kuru (https://www.cdc.gov/prions/index.html; https://www.niaid.nih.gov/diseases-conditions/prion-research; last accessed on 30 August 2022). All known PrDs are progressive, transmissible, have no known effective treatment or cure, are always fatal, and are characterized by the insidious onset of neurological deficits, which are caused in part by the accumulation and aggregation of a misfolded prion protein ‘scrapie’ isoform (PrP**^sc^**) derived from a native cellular prion protein (PrP**^c^**). The rapid development of a progressive systemic inflammation and protein aggregation is very similar in its presentation to Alzheimer’s disease (AD) and other protein-mis-folding disorders such as the tauopathies [20,21,41,42,43,50]. 

Because of their unusual and atypical infective nature, PrDs and prion neurobiology have been intensively studied in considerable detail [19,20,47,48,49,50,51,52,53,54,55,56,57,58,59]. The brain- and CNS-abundant cellular prion sialoglycoprotein PrP**^c^** monomer consists of a constitutively expressed ~209-amino-acid, ~200 kDa glycosylated polypeptide containing a predominant internal α-helical region. This structural region appears to be involved in neuritogenesis, neuronal homeostasis, cell signaling, cell–cell adhesion and interaction, and intercellular communication, and it may provide a protective role against multiple forms of induced physiological stress [19,20,51]. The misfolded isoform of PrP**^c^**, known as PrP**^sc^**, is enriched in pathological β-pleated sheet structure and self-associates into protease-resistant, pro-inflammatory aggregates that are insoluble in most detergent and chaotropic agents [11,19,47]. The molecular mechanisms of PrP**^sc^** neurotoxicity that drive the initiation, development, and progression of PrD appear to be dependent on the unnatural folding associated with the PrP**^c^**-PrP**^sc^** transition and are related to prion aggregation, increased oxidative stress, and the chronic inflammation linked to PrD initiation, maintenance, and progression [19,20,52]. Typically, activated microglia accumulate around PrP**^sc^** aggregates and release cytokines such as IL-1β that play important roles in the inflammatory pathogenesis of PrD and the ***cytokine storm syndrome*** [19,20,47,51,59]. PrDs such as CJD appear to be initiated and driven by the accumulation of abnormally folded, protease-resistant isoforms of PrP**^sc^**, which leads to neuropathologic spongiform changes that coincide with neuroinflammation, microglial activation, and an irreversible and fatal pro-inflammatory neurodegeneration [52,53,54,55,56,57,58,59]. 

## 4. SARS-CoV-2 and PrD: Overlapping Neuropathology

It is of a rather serious current concern that in association with SARS-CoV-2 infection, there are emerging case reports of COVID-19 patients developing PrD and/or are experiencing an acceleration or exacerbation in the development or propagation of this pre-existing, fatal, preclinical and/or already-established, age-dependent neurodegenerative disorder [48,53,56]. A number of interesting and fascinating associations have recently been made between SARS-CoV-2 infection, prion neurobiology, and PrD: **(i)** both SARS-CoV-2-mediated neurological complications and PrDs represent variably transmissible, pro-inflammatory diseases of the brain and CNS, involve a significant disruption in cytokine signaling patterns that is sometimes referred to as the ***cytokine storm syndrome***, and are neurodegenerative, consistently neuro-disruptive, and/or lethal neurological disorders [58,59]; **(ii)** several recent reports link multiple aspects of the ‘S1’ spike protein structure and function, immunology, and epidemiology with PrD, prion-like spread, and prion neurobiology [44,48,60,61]; **(iii)** ‘S1’ spike proteins contain self-associating ‘prion-like’ domains [43,44,48]; **(iv)** ‘S1’ spike proteins are either bound to the SARS-CoV-2 lipoprotein envelope or are in free monomeric form and these domains also appear to play a role in systemic amyloidogenesis in aggregate ‘seeding’ and/or ‘spreading’, which in turn supports systemic inflammation and the formation of pathogenic pro-inflammatory lesions in the brain and CNS that sustain pro-inflammatory neurodegeneration, neuronal cell death, and/or PrD-type change [3,44,48,59,60]; **(v)** the SARS-CoV-2 ‘S1’ spike protein binds to aggregation-prone glycosaminoglycan heparin and heparin binding protein (HBP), amyloid-beta (Aβ) peptides, α-synuclein, tau and prion proteins, and TDP-43 (TAR DNA binding protein 43, critical for the regulation of the viral gene expression), thus facilitating and/or accelerating the coalescence and aggregation of multiple pathological amyloidogenic proteins in nervous tissues, all of which appear to further contribute to the protein-mis-folding characteristic of PrD infection [3,20,22,49,60]; and **(vi)** variations in the prion-like domains of the ‘S1’ spike protein differ among SARS-CoV-2 variants, thus modulating ‘S1’ affinity for the ACE2R and hence the success of SARS-CoV-2 infectivity [3,6,9,43,44]. Targeting the interaction of the SARS-CoV-2 ‘S1’ spike protein with this series of brain-enriched pathological and pro-inflammatory proteins may be a useful therapeutic strategy to reduce aggregation processes, with the aim of limiting progression of the neurodegenerative disease process in COVID-19 patients [11,23,60]. 

## 5. SARS-CoV-2 and Prion Disease (PrD): Case Reports

Multiple case reports of COVID-19 patients developing PrD and/or presenting with exacerbated neuropathological consequences of PrD as a result of SARS-CoV-2 infection have recently appeared in the peer-reviewed scientific literature. Young et al. described a ~60-year-old male patient whose initial indications for CJD occurred in tandem with a fully symptomatic onset of COVID-19, and who deceased 2 months after the first symptom onset. A brain MRI and a comprehensive biofluid analysis revealed an abnormality in inflammatory biomarkers and a SARS-CoV-2-mediated hastening of CJD pathogenesis, which suggest an association between host immune responses to SARS-CoV-2 and an acceleration of the pro-inflammatory neurodegeneration characteristic of idiopathic CJD [53]. Kuvandik et al. described an ~82-year-old female patient whose CJD course was significantly amplified after SARS-CoV-2 infection and COVID-19 vaccination. It was concluded that the role of viral-mediated inflammation and immunity-related conditions for CJD played a significant role in PrD proliferation [55]. Ciolac et al. described SARS-CoV-2 infection in a ~60-year-old female CJD patient who presented with cognitive impairment, gait ataxia, temporo-spatial disorientation, bradykinesia, and multifocal myoclonus and later developed severe COVID-19. The conclusions from this study were that SARS-CoV-2-associated systemic immune responses aggravated the clinical course in patients with CJD, and that systemic inflammation and the host immune responses associated with SARS-CoV-2 appeared to accelerate the rate of neurodegeneration in CJD patients. These studies provide further evidence of the age-dependent neurological effects of SARS-CoV-2 that predisposes vulnerable individuals to an increased progression of CJD [56]. Bernardini et al. recently described a previously healthy ~40-year-old male COVID-19 patient who developed fatal CJD two months after the COVID-19 onset while presenting symptoms of ataxia, diffuse myoclonus, dizziness and loss of coordination, hallucinations and visuospatial deficits. Again, this case study concluded that the brief interval between SARS-CoV-2 respiratory symptoms and CJD neurological symptoms was indicative of a causal relationship between a COVID-mediated inflammatory state, protein mis-folding, and the subsequent aggregation of PrP**^c^** into PrP**^sc^**, again emphasizing the role of SARS-CoV-2 as an significant ‘viral initiator of progressive neurodegenerative disease’ [57]. Olivo et al. described the case of a ~70-year-old male with seizures and a rapidly evolving CJD during an acquired SARS-CoV-2 co-infection, again supporting the notion that CJD during SARS-CoV-2 infection is characterized by an accelerated progression of PrD [58,59]. Taken together, the concept that SARS-CoV-2 worsens, aggravates, or intensifies CJD is deemed urgent because COVID-19 infection foreshadows a disproportionately worse outcome in the elderly who are already at risk of PrD and other forms of age-related neurodegenerative disease [48,53,57,58,59,60,61,62]. These evolving clinically and molecularly evidenced associations between SARS-CoV-2 and CJD underscore an overlapping pathological link between COVID-19 and PrD, both involving systemic inflammation, progressive lethal neurodegeneration, and the potential acceleration of PrD-like protein spread, especially in elderly persons who already possess neurological symptoms. Severe neuroinflammatory reactions and aging are two shared links between neurodegenerative diseases and COVID-19; therefore, COVID-19 patients that have a very high viral load may be at the highest risk of developing long-term adverse neurological consequences [13,48,63,64]. 

## 6. SARS-CoV-2 Infection, PrD, and a Pathological microRNA (miRNA) Signature

Of considerable interest is the effect of SARS-CoV-2 invasion on the molecular genetics and gene expression patterns of the newly infected host, and this is reflected in part by the consequences of SARS-CoV-2 infection on the abundance, speciation, and complexity of a small family of brain-enriched pathology-associated microRNAs (miRNAs). Very recent evidence suggests that dietary polyphenolic compounds may protect against SARS-CoV-2 invasion by modulating the patterns of expression of host cell miRNA [13,64,65]. These ~22-nucleotide single-stranded RNAs (ssRNAs) include several inducible pro-inflammatory miRNAs such as miRNA-146a-5p, miRNA-155-5p, and several others [13,66,67,68,69,70]. A considerable amount of work has focused on the brain-enriched pathogenic miRNA-146a-5p, which was found to be significantly up-regulated in at least 12 categories of PrD in rodents, ruminants, and humans, and after infection by at least 18 neurotropic DNA and/or RNA viruses, including SARS-CoV-2, which infect the human brain, the CNS as well as the immune, lymphatic, hepatic, respiratory, and/or circulatory systems [68,69,70,71]. There is additional evidence that the ACE2R recognized by the SARS-CoV-2 ‘S1’ spike protein is up-regulated by miRNA-146a, and that the many types of PrD and viral infections that induce miRNA-146a-5p and/or miRNA-155 are all associated with advancing systemic inflammation and specific neurological disease symptoms and/or syndromes that are progressive, age-related, insidious, incapacitating, and invariably fatal [13,15,70]. Despite the absence of detectable nucleic acids in prions, both DNA- and RNA-containing viruses, along with prions, significantly and progressively induce miRNA-146a and/or miRNA-155 in the infected host, but whether this represents a reaction to the host’s innate immune response or adaptive immunity, or if it represents a mechanism that enables the invading prion or virus to achieve a successful infection is not well-understood [13,19,20,42,69,70,71]. It is clear, however, that miRNA-146a and/or miRNA-155 signaling underlies several common pathological molecular genetic mechanisms in each of these progressive age-related neurological disorders, and for which there are currently no effective clinical treatments or cures.

## 7. Summary

Since the first cases of SARS-CoV-2 viral infection were reported in Wuhan, Hubei Province, China in early 2020, multiple clinical and epidemiological investigations have yielded complex details concerning the etiopathology and the extraordinary neurological sequelae over the post-COVID-19 period. While the ***long-term*** neurological consequences of SARS-CoV-2 invasion continue to be documented, the ***very long-term*** consequences of SARS-CoV-2 (greater than 2.6 years post-infection) remain unknown. It appears that elderly COVID-19 patients with pre-existing neurological conditions constitute an extremely high-risk category prone to more severe complications of SARS-CoV-2 infection and ***long COVID***. The pathological and neurodegenerative pathways utilized by both SARS-CoV-2 and PrD have been shown to overlap, and this is of increasing concern. There is also the evolving realization that SARS-CoV-2 accelerates and/or intensifies the pathomechanism of PrD and other types of progressive pro-inflammatory neurodegeneration, including AD. This is coupled with the growing recognition of significant cytokine-facilitated immune and pro-inflammatory responses to systemic SARS-CoV-2 invasion and a general viral-mediated hastening of the neurodegenerative disease process [53,54,59,63,72,73,74]. Cytokine-directed therapies may have some benefit in the clinical management of either COVID-19 or PrD [59,63]. Future investigation of the expanding global incidence of COVID-19 variants and the increased occurrence of SARS-CoV-2-modulated neurodegenerative disorders: **(i)** should further unravel the complex neurobiology and inflammatory neuropathology of these pathologically interrelated and overlapping neurological syndromes; and **(ii)** will advance our mechanistic understanding of COVID-19 onset, epidemiology, and propagation, with the long-term goal of expanding and improving our therapeutic treatment strategies for a neurological healthcare crisis that represents one of the worst pandemics in recorded human history.

## Data Availability

All data in this Communication have been derived from the quoted References freely- and openly accessible at the National Institutes of Health (NIH) National Library of Medicine at PubMed.gov (https://pubmed.ncbi.nlm.nih.gov; last accessed on 30 August 2022).

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
