# Peer review of "SARS-CoV-2 Invasion and Pathological Links to Prion Disease"

_biomolecules, 2022, doi:10.3390/biom12091253_

Round 1
Reviewer 1 Report
The commentary discusses the pathological links and similarities of SARS-CoV-2 infection and PrDs by evaluating peer-reviewed studies and whether chronic viral mediated changes can contribute to progressive neurodegenerative disorders, such as PrDs. This commentary is on a timely, important topic, it is informative and written clearly. I recommend this paper to be accepted after minor revisions. I suspect will become increasingly relevant on multiple fronts.
Introduction and other sections:
The use of ‘see below’ – would it be better to refer to ‘Section x’?
Section 2, SARS-CoV-2 structure, function and neuroinvasion:
Line 99: ‘Prion like attributes’ – could these be explained in more detail?
Section 3, PrD and prion neurobiology:
Line 113 and 114: should be PrPSc and PrPC
I am wondering whether it is worth including information about onset age and duration for sporadic genetic forms of PrD and incubation times for acquired forms of PrD.
Section 4, SARS-CoV-2 and PrD – overlapping pathological neurobiology:
Line 143: references only 3 case reports, but 5 are discussed in the following sections. Are reference 57 and 58 missing for a specific reason?
Line 145/146: ‘highly transmissible’ – I would argue that PrDs are not highly transmissible and that the mechansims of transmission between COVID and PrD are very different. I am not sure I would use this point to illustrate a similarity between COVID and PrD.
Line 147: Is the dysregulation of cytokine signaling in PrD a ‘cytokine storm syndrome’? Is there a reference for this statement?
Line 149: ‘multiple aspects of the S1 spike protein’ – what are these links/similarities?
Last sentence: I wonder, would the disrupted cytokine signaling also provide pointers to future therapeutic strategies for PrD?
Section 4, SARS-CoV-2 and PrD case reports:
Line 205-209: Is this statement too strong? Are these true links or similar/identical pathways?
General comments:
1. Incubation times of prion diseases are generally long (years/decade(s)). One would expect the effects of COVID-19 to be seen over a much longer timeframe.
2. PRNP mutation carriers, who are likely to develop PrD in their lifetime, would be an important cohort to follow and establish their COVID history.
3. Are 5 isolated case studies enough to draw any conclusions on the causality of COVID, leading to or exacerbating the course of PrD? In 600 million COVID infected people and PrD rates of 1-2 per million people per year, one expects cases of concomitant COVID/PrD illness. Surveillance units should monitor closely whether the rate of PrD is increased in the general population and cohorts of severe or long COVID, also whether the age of onset and duration of illness have changed in PrD from before the COVID pandemic.
Reviewer 2 Report
Dear the editor and the staff,
I believe it is very good, which makes the statement about the relationship between coronaviruses and prion diseases. However, I think we need to define the term tightly with regard to cytokine storms. Cytokine storm, as one might imagine from the word "storm," is an uncontrolled inflammatory response that occurs when the immune system releases too many cytokines. In a cytokine storm, cytokines are released in excess. As a result, various types of immune cells, including T cells, macrophages, and natural killer cells, are over-activated. The uncontrolled activity of these cells is known to lead to tissue damage, organ dysfunction, and sometimes death. However, the cytokine storm that occurs in prion diseases is localized in the brain and is not a systemic response. I think we should define cytokine storms in this paper by stating that they are different from cytokine storms as usually defined.
